# Application of Explainable Artificial Intelligence Based on Visual Explanation in Digestive Endoscopy

**DOI:** 10.3390/bioengineering12101058

**Published:** 2025-09-30

**Authors:** Xiaohan Cai, Zexin Zhang, Siqi Zhao, Wentian Liu, Xiaofei Fan

**Affiliations:** Department of Gastroenterology, Tianjin Medical University General Hospital, No. 154, Anshan Road, Heping District, Tianjin 300052, China; caixiaohan333@163.com (X.C.); 15504768589@163.com (Z.Z.); 17746161449@163.com (S.Z.)

**Keywords:** explainable artificial intelligence, machine learning, deep learning, visual explanation, digestive endoscopy, medical imaging

## Abstract

At present, artificial intelligence (AI) has shown significant potential in digestive endoscopy image analysis, serving as a powerful auxiliary tool for the accurate diagnosis and treatment of gastrointestinal diseases. However, mainstream models represented by deep learning are often characterized as complex “black boxes,” with decision-making processes that are difficult for humans to interpret. The lack of interpretability undermines physicians’ trust in model results and hinders the broader use of models in clinical practice. To address this core challenge, Explainable AI (XAI) has emerged to enhance the transparency of decision-making, thereby establishing a foundation of trust for human–machine collaboration. The review systematically reviews 34 articles (7 articles in esophagogastroduodenoscopy, 13 articles in colonoscopy, 9 articles in endoscopic ultrasonography, and 5 articles in wireless capsule endoscopy), focusing on the research progress and applications of XAI in the field of digestive endoscopic image analysis, with particular emphasis on the visual explanation-based methods. We first clarify the definition and mainstream classification of XAI, then introduce the principles and characteristics of key XAI methods based on visual explanation. Subsequently, we review the applications of these methods in digestive endoscopy image analysis. Lastly, we explore the obstacles presently faced in this domain and the future directions. This study provides a theoretical basis for constructing a trustworthy and transparent AI-assisted digestive endoscopy diagnosis and treatment system and promotes the implementation and application of XAI in clinical practice.

## 1. Introduction

The continuous development of artificial intelligence (AI) is currently driving significant changes in medicine. It can enhance medical clinical decision-making, reduce medical errors, and improve patient prognosis [1]. Machine Learning (ML) is a subset of AI. Its core lies in constructing a mathematical model through algorithms and optimizing the model’s parameters using data, thereby enabling it to make accurate predictions or decisions on unknown data [2,3]. However, traditional ML models often require manual feature extraction, which limits its application in more complex tasks. Deep learning (DL) is a specialized area within ML. Its fundamental structure relies on multi-layer artificial neural networks that mimic neuron activity transmission in the human brain to process information. It can independently learn features from input data and shows significant advantages in processing complex data, enabling it to handle more challenging tasks, such as convolutional neural networks (CNNs), recurrent neural networks (RNNs), and so on [4]. Currently, AI is increasingly being used in medical imaging. By training on large amounts of medical image data, it enables the automatic detection and diagnosis of lesions, which helps clinicians make faster and more informed decisions and improves outcomes for patients [5]. In the field of digestive endoscopy, AI-related computer-aided detection (CADe) and computer-aided diagnosis (CADx) have been widely studied. By processing and analyzing vast amounts of digestive endoscopic image data, it is expected to overcome the limitations in endoscopic examinations, thereby improving the quality of clinical diagnosis [6,7]. However, despite the popularity of these AI models, there are still concerns or doubts about their clinical applications. One of the main challenges we face is the inherent “black box” nature of some AI models, especially in complex DL algorithms. DL models contain multiple hidden layers, making it very challenging for humans to comprehend how AI models reach their final conclusions [8].

However, in the field of healthcare, AI-driven medical decisions are closely related to human life and death. Inaccurate or incorrect results can have serious consequences for patients. Therefore, it is essential to provide explanations for AI results to ensure that diagnostic outcomes align with medical expertise, thereby earning the trust of doctors, patients, and regulators. Meanwhile, newer regulations like the European Union’s General Data Protection Regulation (GDPR) have made the use of black-box models more challenging in all sectors, including healthcare, as they require traceability of decisions [9,10]. Therefore, while achieving high performance in AI, we believe that research into AI explainability methods is crucial, which has resulted in the rise in explainable AI (XAI) [10]. XAI can enhance the interpretation of the outputs of black-box models and ensure the credibility of the results.

Van der Velden et al. [11] included 223 relevant studies to comprehensively analyze the application of XAI methods in medical image analysis. They discussed and compared the advantages and disadvantages of various XAI methods and put forward their unique insights into the future development of XAI. The results showed that visual explanation is the most commonly used interpretability method in medical image analysis. Salih et al. [12] included 32 articles to conduct a retrospective analysis of the application of XAI in various cardiac imaging methods. Among them, the visual explanation method achieves model interpretability by visualizing relevant features, accounting for approximately half of the article. Similarly, Qian et al. [13] included 56 articles to study the application of XAI in the analysis of MRI images of different human body parts, and identified the evaluation metrics for XAI methods. The results showed that XAI based on visual explanation accounted for more than 60%. The above studies highlight the widespread clinical application of XAI based on visual explanation. However, there are still relatively few reviews on XAI, especially XAI based on visual explanation, in the field of digestive endoscopic image analysis.

Therefore, based on the above background, this review aims to systematically summarize the current research status and application progress of XAI in the field of digestive endoscopy, with a focus on XAI centered on visual explanation. At the same time, this paper will also analyze the clinical application value, existing challenges, and future development directions in this field, so as to provide theoretical basis and practical guidance for promoting the standardized application of XAI in digestive endoscopy diagnosis and treatment. In Section 2, we define XAI and outline its general classification methods. In Section 3, we elaborate on the principles, advantages, and disadvantages of XAI methods based on visual explanation. In Section 4, we review the specific applications of XAI methods based on visual explanation in the analysis of digestive endoscopy images. In the end, we summarize the benefits, challenges, and future development trends of XAI in clinical practice to promote its broader implementation.

## 2. Definition and Classification of XAI

### 2.1. Definition of XAI

Technically speaking, the academic community has not yet formed a unified and clear definition of XAI [10]. The terms “interpretability” and “explainability” are frequently used interchangeably in relevant research, and there are still divergences in the academic community regarding the definition of their concepts [14]. Among them, Rudin believes that AI interpretability refers to the characteristics that the model itself can be directly understood by humans; while AI explainability emphasizes clarifying the working mechanism of the original black box model by constructing an auxiliary explanation model [15]. In this paper, we will use the term “explainable artificial intelligence”, which refers to the ability to generate details or reasons to explain how black-box models make decisions, thereby making the results of AI more understandable to humans [16,17].

### 2.2. Classification of XAI

XAI can be categorized into the following three types based on different classification standards: model-based explanation versus post hoc explanation, model-specific explanation versus model-agnostic explanation, and global explanation versus local explanation (see Figure 1).

#### 2.2.1. Model-Based Explanation vs. Post Hoc Explanation

Model-based explanation involves constructing relatively simple models that allow humans to use their domain knowledge to understand how the model converts inputs into outputs, like traditional linear regression and decision trees. These models not only provide model prediction results but also feature importance scores. Post hoc explanation refers to analyzing a constructed model (such as a CNN model) to gain insights into the model’s learning relationships [18]. The difference between it and model-based explanation lies in that this method attempts to train a new model to explain the black-box network; whereas model-based explanation mandates that the model itself is interpretable [11]. For instance, Gradient-weighted Class Activation Mapping (Grad-CAM) is a common post hoc interpretability method applied in DL models.

#### 2.2.2. Model-Specific Explanation vs. Model-Agnostic Explanation

Model-specific explanation is closely related to the internal structure of a particular model and can only be applied to specific models or algorithms, which to a certain extent restricts the freedom of model selection [9]. For example, the Graph Neural Network Explainer (GNN Explainer), as a specialized interpretability tool, provides interpretive analysis only for the predictions of graph neural network models [19]. In contrast, model-agnostic explanations remain unaffected by which model is selected and do not require probing the internal working mechanisms or parameters of the model. They directly operate on the model’s inputs and outputs, observing how modifications to inputs influence outputs to discover certain regions related to the outputs [11].

#### 2.2.3. Global Explanation vs. Local Explanation

Global explanations provide the general learned relationships of a model, identifying features that contribute to classifying a certain target across all instances. They reveal the general decision-making patterns of the model. In contrast, local explanations present the input-output relationships for individual instances, i.e., the features or attributes that are predicted to influence the specific target output [11,20]. For example, SHapley Additive exPlanations (SHAP) analysis can not only provide global explanations to reveal the ranking of important features related to overall prediction results, but also provide important features associated with the result of individual sample to provide local explanations [21].

## 3. XAI Methods Based on Visual Explanation

Currently, XAI methods applied in medical image analysis are generally classified into three categories: visual explanation, textual explanation, and case-based explanation [11]. Among them, visual explanation is the most prevalent in medical image analysis [4]. It mainly analyzes which parts of the original image have affected the final output result through backpropagation or perturbation methods [9]. Most of them are displayed in the form of heatmaps, where different colors in the heatmap indicate the degree of influence of that part on the prediction result [22], and generally provide post hoc explanation (Figure 2). Textual explanation refers to providing descriptive text to explain predictions, such as visual question answering tasks, image captions, image captions with visual explanation [23]. It can establish a connection between medical images and semantic information. Case-based explanation makes predictions by analyzing examples or data related to the current task, such as Testing with Concept Activation Vectors (TCAV) [24]. Current research indicates that XAI methods based on visual explanation occupy a dominant position within the domain of analyzing medical images. They display features related to predictions on images in an intuitive and comprehensible manner. Meanwhile, the plug-and-play, model-agnostic characteristics and the availability of open-source implementations further facilitate their widespread adoption [4]. Therefore, this paper mainly reviews the application of XAI methods based on visual explanation in medical images, especially in endoscopic images.

### 3.1. Backpropagation-Based Methods

Backpropagation-based methods estimate the impact of gradients, weights, and activations by performing one or more forward propagations in the network and calculating partial derivatives during the backpropagation phase to generate attribution maps [25]. They operate relatively quickly, but the relationship between the results and output changes is weak [9]. The following are several key methods based on backpropagation.

#### 3.1.1. Saliency Map Visualization

Simonyan et al. [26] first proposed a method using backpropagation called saliency map visualization. It calculates gradients through backpropagation and uses the gradients to highlight the correlation between input pixels and prediction results, thereby achieving model visualization [20].

#### 3.1.2. Deconvolution Networks (DeconvNets) and Guided BackPropagation (GBP)

Deconvolution networks (DeconvNets) consist of a series of deconvolution and unpooling layers and generate attribution maps by setting negative gradients to zero during the backward pass to visualize the neural activations of each layer [20,27]. Guided BackPropagation (GBP) is an improvement over the deconvolution method. It visualizes gradients with respect to the image while performing backpropagation through the ReLU activation function. By guiding backpropagation, it suppresses negative gradients from coming back and underscores the pixels that most significantly influence the output, generating a saliency map [13,20]. While both DeconvNets and GBP reveal the fine-grained details in the image, they have limitations: their visualizations cannot distinguish different categories, as the saliency maps for various classes often look very similar [28].

#### 3.1.3. Class Activation Mapping (CAM)

Proposed by Zhou et al. [29], Class Activation Mapping (CAM) is a method designed to interpret CNN models. First, it implements a global average pooling operation in the terminal convolutional layer of the network to replace the fully connected structure, and then projects the output layer’s weights back to the feature mapping space, thereby generating a heat map to realize the identification and visualization of important regions in the image.

#### 3.1.4. Gradient-Weighted Class Activation Mapping (Grad-CAM)

Gradient-Weighted Class Activation Mapping (Grad-CAM) is an extension and generalization of CAM. The difference between it and CAM is that it is applicable to various types of CNNs, while CAM can only be used in CNNs with global average pooling [13]. Meanwhile, Selvaraju et al. [28] also proposed Guided Grad-CAM, which combines Grad-CAM with guided backpropagation. This hybrid approach generates high-resolution, fine-grained visualization maps that not only locate the relevant regions but also provide detailed insight into the features influencing the model’s decision, uniting the precise localization capability of Grad-CAM with the high-resolution attributes of guided backpropagation.

#### 3.1.5. Layer-Wise Relevance Propagation (LRP)

Bach et al. [30] first proposed layer-wise relevance propagation (LRP) to realize the pixel-wise explanation of the decisions made by nonlinear classifiers. LRP attributes relevance scores to individual input features to explain the predictions of neural networks. This method starts from the output layer of the network and traces back layer by layer to the input layer through backpropagation. During the iteration process of each layer, the algorithm assigns corresponding contribution scores to each neuron in the previous layer and strictly follows the principle of relevance conservation to ensure that the overall correlation strength remains consistent during the transmission process [30,31]. This process is used to show the impact of features on the prediction results.

### 3.2. Perturbation-Based Methods

Perturbation-based methods can be executed by altering, eliminating, or concealing specific input features and measuring the differences from the original output. Among them, the features that have the greatest impact on the output results are considered the most important [9]. Since this process does not require access to the internal structure of the model, this method is classified as a model-agnostic explanation. However, since this method needs to make predictions on multiple inputs and outputs, it generally takes longer than backpropagation-based methods [32].

#### 3.2.1. Occlusion

The occlusion predicts the impact of relevant regions on output results by altering input images (features) [4]. Among them, the occluded parts that have a greater impact on the output results are considered to be of high importance, while those have a smaller impact on the output results are assigned low importance [11,33].

#### 3.2.2. Local Interpretable Model-Agnostic Explanations (LIME)

Ribeiro et al. [34] introduced local interpretable model-agnostic explanations (LIME), a method that gives explanations through training a relatively simple model to simulate a complex model, thereby learning the relationship between perturbed input data and output changes. For example, it uses a linear model to approximate and replace a CNN. It utilizes the resemblance between the modified input and the initial input as weights to make sure that the explanations given by the simple model with heavily modified inputs are more trustworthy [11,17]. In image analysis, LIME can highlight the image regions that make the most significant contributions to specific class decisions, thereby achieving local interpretation of the model.

#### 3.2.3. SHapley Additive exPlanations (SHAP)

Lundberg and Lee [35] introduced the concept of SHapley Additive exPlanations (SHAP), which utilizes the principle of Shapley value in cooperative game theory to provide a basis for the prediction results of ML models. This method estimates the marginal contribution of each feature to the model’s prediction results by sampling different feature combinations multiple times and evaluating the changes in their outputs [9,11]. Meanwhile, after pixelating the Shapley values, different colors can be used to indicate the active or passive impacts of different characteristics on the model’s decision-making. This method can intuitively display the importance ranking of various features, helping to interpret the model’s decision-making process comprehensively [11,36].

## 4. Applications of XAI Methods Based on Visual Explanation in Digestive Endoscopy

We searched articles published in the PubMed database from 2015 to 2025 to identify the current applications of XAI based on visual explanation in the analysis of digestive endoscopic images. The search terms are as follows: (explainable artificial intelligence” OR “interpretable artificial intelligence” OR “artificial intelligence” OR “AI” OR “machine learning” OR “deep learning” OR “convolutional neural network”) AND (endoscopy). The inclusion criteria are as follows: (1) The XAI models are applied to digestive endoscopic image analysis; (2) The article adopts specific visual explanation methods to explore the interpretability of AI models; (3) Complete article information is accessible. We also included key articles identified through manual searching or reference lists. Then, the above articles were manually screened and evaluated again, and a total of 34 eligible articles were finally included.

### 4.1. Applications in Esophagogastroduodenoscopy

Esophagogastroduodenoscopy (EGD) is an important examination method for diagnosing upper gastrointestinal diseases, especially for esophageal and gastric diseases. It allows direct visualization of the mucosal surface, facilitating the precise diagnosis and assessment of illnesses. Among them, white light endoscopy (WLE) is the most widely used examination method, mainly used to observe and evaluate the gross morphological appearance of lesions. Compared with WLE, chromoendoscopy (CE) and narrow-band imaging (NBI) can make the scope and outline of lesions clearer, thereby improving the ability to identify lesions. Magnifying endoscopy (ME) combines the functions of endoscopy and microscopic imaging, enabling magnified observation of the microstructures of the mucosal surface such as glandular openings and microvascular, thus demonstrating unique diagnostic advantages in the identification of early-stage cancers. It has unique diagnostic value in detecting early-stage cancers. Clinically, endoscopists often combine ME with CE or NBI; upon suspicious findings in WLE, they first use staining or NBI mode to highlight the outline of the lesion tissue and then switch to the magnifying mode to observe the fine structure of the local mucosa. This strategy can improve the detection rate of lesions and the accuracy of biopsy [37]. In addition, with technological advancements, more and more AI and XAI are integrated into EGD to support clinical decision-making. These systems can highlight areas of concern, assist in classifying lesions, and provide visual explanations to endoscopists, enhancing diagnostic reliability and precision [38]. Table 1 summarizes XAI applications and visual explanation methods used in EGD.

Various XAI methods are used in EGD image analysis to assist in clinical disease diagnosis. Regarding the identification of esophageal-related diseases, gastroesophageal reflux disease (GERD) is a common digestive system disease disorder. It can be diagnosed through endoscopic examination, which reveals characteristic mucosal damage, or through reflux testing that detects abnormal esophageal acid exposure [46]. Ge et al. [39] utilized 2081 endoscopic images to establish a deep learning model based on DenseNet-121 for identifying the Los Angeles classification (LA-grade) of GERD. The model achieved an area under the curve (AUC) of 0.968, and its classification accuracy (86.7%) was significantly higher than that of junior endoscopists (71.5%) and senior endoscopists (77.4%). Meanwhile, heatmaps were generated based on Grad-CAM to solve the black-box problem of the model. Barrett’s esophagus (BE) is a disease in which the mucosal cells of the lower esophagus undergo significant changes, which can progress to esophageal adenocarcinoma in severe cases. De Souza et al. [40] constructed various CNN models for the identification of BE and esophageal adenocarcinoma, and adopted multiple XAI tools such as saliency map and GBP to explain the model decisions. The results showed that the regions highlighted by the saliency map had a high consistency with human segmentation, achieving the best explanation results. This study not only utilized multiple XAI methods but also evaluated different XAI methods by comparing XAI outputs with expert annotations. The histological staging of early squamous cell neoplasia (ESCN) can be predicted by observing the morphological characteristics of its specific microstructure, namely the intraepithelial capillary loops (IPCLs), which are regarded as endoscopic markers of ESCN [47]. Currently, several studies have constructed XAI models to improve the identification of IPCLs. For example, García-Peraza-Herrera et al. [41] used 67,742 video frames from 114 patients to build a CNN for binary classification of IPCLs, achieving an average accuracy of 91.7%. CAM maps were employed to highlight the parts contributing to the results. However, the above-mentioned model can only identify static video frames and cannot run in real-time, which limits their application in clinical practice. Subsequently, Everson et al. [42] used 67,742 high-quality magnifying endoscopy with narrow band imaging (ME-NBI) images from 115 patients to train a CNN to classify IPCLs and predict ESCN histological staging. With an average diagnostic accuracy of 91.7%, the model’s performance closely approximated the comprehensive diagnostic level (94.7%) of endoscopic experts. Crucially, this CNN operated at video rate, possessing the ability of real-time prediction, thus overcoming the temporal limitations of previous static-frame models. Additionally, CAM was used to highlight the features affecting classification prediction outcomes.

In terms of assisting in the identification of stomach-related diseases, multiple studies have constructed XAI models based on CAM to achieve diagnosis of early gastric cancer (EGC) and prediction of gastric cancer invasion depth. Ueyama et al. [43] developed a CAD system using 5574 ME-NBI images to identify EGC and achieved high diagnostic accuracy. The comprehensive performance evaluation metrics of this CNN model yielded excellent results: the area under the curve (AUC) reached 99%, overall accuracy was 98.7%, sensitivity achieved 98%, specificity attained 100%. Finally, Grad-CAM was used to visualize the image regions affecting the classification results, and these regions were consistent with those identified by endoscopists, providing an interpretable analysis of the model. However, although the model achieved excellent results, the study was a single-center study and lacks external validation, limiting the generalizability of the findings. Hu et al. [44] constructed an AI model (EGCM) for identifying EGC using 1777 ME-NBI images from 3 centers and conducted a human–machine comparison. The results showed that the AUC of the model in both internal and external validation sets was as high as approximately 80%. Meanwhile, the diagnostic performance of the model (with an accuracy of 0.77) was similar to that of senior experts (with an accuracy of 0.755) and better than that of junior experts (with an accuracy of 0.728). The diagnostic performance of all doctors was improved with the assistance of EGCM, and Grad-CAM highlighted the abnormal areas of the lesions. Accurate assessment of the invasion depth of gastric cancer is critical for selecting the optimal treatment strategy. Cho et al. [45] trained two CNN models using WLE images of gastric cancer patients to classify gastric tumors as either confined to the mucosal layer or demonstrating submucosal invasion, and validated them in external datasets. Among them, the DenseNet-161 model performed better, with an AUC as high as 0.887 in both internal and external validation sets. CAM effectively displayed the characteristic regions of the tumor to explain the model.

### 4.2. Applications in Colonoscopy

Colorectal cancer (CRC) is the fourth leading cause of death among malignant tumors worldwide [48]. The 5-year survival rate can reach 91% in the early stage, but drops to approximately 14% in the advanced stage [49]. Most of them follow the adenoma-carcinoma progression sequence [50]. Therefore, timely detection and treatment of polyps and/or adenomas through colonoscopy can significantly decrease the incidence rate of colorectal malignant tumors. However, the accuracy of polyps and/or adenomas detection is affected by factors such as manual fatigue and physician experience. Studies have shown that the missed diagnosis rate of adenomas in serial colonoscopy is 26% [51,52]. Meanwhile, the removal of polyps and/or adenomas must take into account the increased medical costs, including those related to pathological examinations [53]. Therefore, comprehensive identification of lesions and accurate optical diagnosis can significantly reduce costs. In this context, XAI can be used to assist physicians in performing colonoscopies, improving the quality of detection while providing physicians with diagnostic insights. Table 2 summarizes XAI applications and visual explanation methods used in colonoscopy.

In terms of assisting in the identification of colorectal polyps, various different XAI models have been developed so far and have achieved excellent results. Chen et al. [54] collected 4189 colonoscopic images containing polyps, cecum, and different levels of bowel cleanliness to train models based on CNN and Transformer. These models are used to intelligently evaluate key quality indicators of colonoscopy. The results show that the EfficientNetB2 model exhibited excellent performance on both the validation set and the test set. Grad-CAM, Guided Grad-CAM, and SHAP methods revealed the regions that have an impact on the prediction results. Wickstrom et al. [55] used three network architectures to realize the semantic segmentation of colorectal polyps and achieve polyp identification at the pixel level. Meanwhile, GBP was used to highlight the important features for polyp prediction, indicating that the model leverages the contours and morphological characteristics of polyps for its predictive analysis. However, the aforementioned models only identify polyps and do not distinguish between different types of polyps. Since hyperplastic polyps rarely undergo malignant transformation, unnecessary endoscopic resection can increase medical costs without added benefit. Therefore, accurate diagnosis of polyp types is important to avoid inappropriate resection. In this context, Jin et al. [56] collected images of 1100 small adenomatous polyps and 1050 small hyperplastic polyps under NBI to train a CNN model for further distinguishing whether colorectal polyps are adenomatous or hyperplastic. The results of the study showed that the accuracy of the model in differentiation was 86.7% (95%CI: 82.3–90.3). Moreover, with the help of AI, the overall diagnostic accuracy of endoscopists was significantly improved (from 82.5% to 88.5%, *p* < 0.05), and the overall diagnostic time was significantly shortened (from 3.92 s to 3.37 s, *p* < 0.05). Meanwhile this study generated heatmaps using Grad-CAM methods and visualized the highlights that overlaid the polyp, which can aid the endoscopist to accept AI insights. However, this model was only validated in a single center, so its generalization ability in clinical practice may be unsatisfactory. Although the risk of small polyps progressing to CRC is very low, those with advanced features still pose a high risk [67]. Therefore, to further identify colorectal polyps with advanced features, Zhang et al. [57] constructed a CNN model using NBI images to classify colorectal polyps as either with or without advanced features. The model output included NBI images with Grad-CAM heatmaps to highlight the interpretability of the model. The accuracy of the model in the internal and external validation sets was 0.880 (0.839–0.916) and 0.870 (0.843–0.896), respectively, with AUC values of 0.942 (0.847–0.961) and 0.926 (0.846–0.946), respectively, indicating good robustness. Compared with 19 endoscopists, the AI model demonstrated a greater diagnostic performance (*p* < 0.05). Furthermore, in the prospective test, compared with the non-assisted group, the endoscopists in the AI-assisted group identified more polyps with advanced features and showed higher accuracy, sensitivity, specificity, positive predictive value (PPV), and negative predictive value (NPV) (all *p* < 0.001). The study achieved excellent results in the internal and external validation sets as well as the prospective test set, demonstrating the model’s strong generalization ability and making the model results more credible. Meanwhile, human–machine comparison further illustrated the model’s auxiliary role for clinicians. However, both studies focus on binary classification of polyps and do not involve multiple classifications of colorectal polyps. Addressing this, Choi et al. [58] used 3000 endoscopic images of colorectal adenomas to construct three CNN models based on Inception-v3, ResNet-50, and DenseNet-161, respectively, and compared the performance of the CNN models with endoscopists with different years of experience. The models aimed to classify images into multiple pathological categories: normal, tubular adenoma with low-grade dysplasia (TALGD), tubular adenoma with high-grade dysplasia (TAHGD), and carcinoma (CA). The results showed that the three CNN models outperformed the endoscopy expert group, and the model based on DenseNet-161 achieved the best performance. CAM was used to highlight the relevant areas of the images, enhancing model interpretability. Although these models have achieved multi-classification of the pathological types of colorectal adenomas, the models’ accuracy in identifying TAHGD and CA lesions is lower than that in identifying normal and TALGD lesions. Future improvements can be made to enhance its performance.

At present, several studies have utilized heatmap-based methods to construct XAI for assisting in the identification of inflammatory bowel disease (IBD). For example, Chierici et al. [59] constructed a DL model to identify ulcerative colitis (UC) and Crohn’s disease (CD). GBP showed that typical endoscopic features of IBD, such as mucosal erythema, appear to have higher attribution values, achieving direct visual interpretation. Sutton et al. [60] utilized images from the HyperKvasir dataset to construct an interpretable CNN model for the diagnosis and grading of UC. The Grad-CAM displayed the image regions used for predicting the output such as fibrin covering ulcers, in the form of a heat map, which was consistent with the pathology of UC. However, results in the form of heatmaps can only provide local explanations and do not provide a comprehensive understanding of the entire model’s decision process. To address this limitation, Weng et al. [61] employed SHAP method to construct an interpretable XGBoost model integrating endoscopic features to distinguish CD from intestinal tuberculosis (ITB), providing both local and global explanations for the model to identify important features.

In addition to the above models, there are currently various ensemble models built based on datasets for identifying gastrointestinal lesions. By constructing ensemble models, the advantages of multiple models are utilized to improve diagnostic performance. Gabralla et al. [62] integrated the output of the pretrained CNN models with a meta-learner (Support Vector Machine, SVM) to construct a stacking model (Stacking-SVM) for colon cancer identification. The results showed that compared with the CNN models, the Stacking-SVM model achieved the highest performance in two different datasets. Additionally, Grad-CAM generated heatmaps to visualize the impact of different regions on the prediction results. Binzagr F et al. [63] developed an ensemble model comprising 3 CNN models-InceptionV3, InceptionResNetV2, and VGG16-based on the KvasirV2 dataset to identify polyps, UC, and esophagitis. The hybrid architecture yielded a classification accuracy of 93.17%, with its F1 score reaching 97%. SHAP was used to provide explanations for the model’s predictions to doctors. Similarly, Auzine et al. [64] also constructed an ensemble model for identifying polyps, UC, and esophagitis based on the aforementioned CNN models. The accuracy of this ensemble model reaches 96.89%. SHAP and LIME methods was used to enhance the comprehensive understanding of the model’s decision-making. Unlike the aforementioned methods for constructing ensemble models, Dahan et al. [65] integrated the deep convolutional neural network model with Swin Transformer to build a hybrid model. This hybrid model is designed to extract more comprehensive endoscopic image features, thereby achieving better identification of gastrointestinal diseases. The accuracy of this hybrid model reaches 93.43%. Grad-CAM highlights the regions that have the most significant impact on the model’s results. Similarly, Gideon et al. [66] utilized the Kvasir and GastroNet datasets to develop a deep learning model incorporating CNNs, RNNs, and the transformers to detect gastroenterological diseases while providing explainability through Grad-CAM and SHAP. The Ensemble model outperforms the individual model showing an accuracy of 92.6%. Grad-CAM heatmaps are used to show which parts of medical images are most relevant to the model predictions. SHAP analysis finds the most important features driving the model’s predictions are texture and color. These interpretability methods help doctors interpret model decisions and build trust in their predictions. In conclusion, the accuracy rates of the above five ensemble models all exceed 90%, suggesting that ensemble models have great development potential in disease diagnosis.

### 4.3. Applications in Endoscopic Ultrasonography

Endoscopic ultrasonography (EUS) integrates ultrasonic imaging and endoscopic visualization functions, providing high-quality ultrasonic images for organs such as the gastrointestinal tract and pancreas. This examination enables high-resolution real-time imaging of the digestive tract’s mucosal architecture, significantly enhancing the detection of characteristics and ranges of lesions [68]. Therefore, it has important application value in the diagnosis of pancreatic diseases, evaluation of submucosal lesions, and determination of the depth of tumor invasion. Compared to conventional ultrasound (US), computed tomography (CT), and magnetic resonance imaging (MRI), EUS may detect smaller lesions [69]. Currently, EUS serves as a vital tool in the diagnosis of diverse gastrointestinal disorders, improving the disease detection rates [70]. However, the diagnostic accuracy of EUS largely depends on the professional knowledge, practical experience and technical level of the operating physicians. The considerable training costs and prolonged learning curve make it hard to master EUS diagnostic skills [71]. Therefore, combining EUS with XAI may help physicians improve the accuracy of clinical diagnosis while enhancing detection efficiency. Table 3 summarizes XAI applications and visual explanation methods used in EUS.

Explainable methods based on Grad-CAM and SHAP are widely used in constructing XAI models to help diagnose diseases related to the pancreas. Gu et al. [72] developed a deep learning radiomics model based on EUS images to effectively identify pancreatic ductal adenocarcinoma (PDAC), and its performance is superior to most clinical experts. Meanwhile, the heatmaps generated by Grad-CAM show that the low to mixed echo areas within the tumor and the tumor boundary areas are of great value to the model’s diagnosis, which helps to understand the diagnostic results. Yi et al. [73] constructed explainable ML models based on DL features extracted from EUS images to distinguish pancreatic neuroendocrine tumors (PNETs) from pancreatic cancer, with Grad-CAM and SHAP clarifying and visualizing the model outputs. However, the above models are all based on static image recognition and have not been validated on dynamic video sets, limiting their further clinical application. Marya et al. [74] addressed this limitation by collecting EUS static images and video information to train a CNN model for distinguishing autoimmune pancreatitis (AIP) from PDAC, chronic pancreatitis (CP), and normal pancreas (NP). This model achieved excellent results in both image datasets and video datasets. Meanwhile, by occluding different pixels, heatmaps are generated to assess the features of the model in identifying AIP and PDAC. The heatmap analysis shows that patients with AIP have significantly more high-scoring subregions in the pancreas than those with PDAC, while PDAC patients have more high-score subregions in the retroperitoneum. Nonetheless, the aforementioned studies mainly focus on the lesion area and ignore the surrounding area of the lesion, which may contain valuable diagnostic information. Therefore, Mo et al. [75] constructed a model based on multilayer perception (MLP) integrating radiomics features from intratumoral and peritumoral regions to predict the pathological grade of PNETs. This study provides insights into the value of peritumoral regions, especially the tumor-adjacent parenchyma, in disease diagnosis. Meanwhile, it uses SHAP values to visualize the importance of features, thereby providing interpretability for the model’s predictions. However, the aforementioned models only utilize single-modal radiomics information and ignore the potential diagnostic impact of other aspects, such as laboratory test results. Integrating multi-modal information may improve the accuracy and robustness of the model’s diagnosis. For instance, Cai et al. [76] constructed a multimodal ML model that combined radiomics features from EUS images of pancreatic lesions and clinical characteristics to identify pancreatic lesions. The findings indicated that the multimodal model performed better. Meanwhile, SHAP method was used to achieve interpretability analysis at both the overall and individual levels (Figure 3). However, the model lacks multi-center external validation and may have overfitting issues. Therefore, Cui and colleagues [77] constructed a multimodal AI model using EUS images and clinical date of patients with pancreatic lesions from multiple centers to distinguish pancreatic cancer from non-cancerous lesions, which effectively solved this problem and greatly improved the credibility of the model’s performance. Meanwhile, this study conducted interpretability analysis using Grad-CAM and SHAP methods and demonstrated that such interpretability analysis improved physicians’ acceptance of AI prediction results.

Since EUS can identify the morphological characteristics of gastrointestinal tumors and the layered architecture of the gastrointestinal wall, it is also applied in evaluating subepithelial lesions and the depth of tumor invasion. Liang et al. [78] collected various clinical and pathological data from patients with gastrointestinal stromal tumors (GISTs), including high-risk features on EUS, to construct a model for predicting the malignant potential of gastric GISTs. SHAP indicated that high-risk features on EUS, tumor size, tumor boundaries, and monocyte-to-lymphocyte ratio are key variables affecting the model results. In terms of evaluating the depth of tumor invasion, Liu et al. [79] constructed a DL model to identify the lesion invasion depth and lesion source of esophageal submucosal tumors. The overall accuracy of the model can reach 82.49%, and CAM identifies the lesion area in the image to improve the interpretability of the model. However, it remains a single-center study with a single source of training data. Subsequently, Uema et al. [80] developed an AI-based EUS system aimed at diagnosing the invasion depth of EGC and collected data from 10 institutions for adequate external validation. The results showed that the AUC of the model in the internal and external validation sets was 87% and 81.5%, respectively, with accuracies of 82.2% and 74.1%, respectively. Notably, in the external validation set, the diagnostic performance of the model was comparable to experts. CAM visualized the regions of interest of the AI model.

### 4.4. Applications in Wireless Capsule Endoscopy

Wireless capsule endoscopy (WCE) is a non-invasive diagnostic technique used for gastrointestinal examinations. After the examinee swallows the capsule, the device can move along with gastrointestinal peristalsis and continuously collect images of the digestive tract mucosa during this process, so as to realize the exploration of the interior of the gastrointestinal tract. It has been proven to be of great value in evaluating focal lesions of the digestive tract, such as gastrointestinal bleeding and ulcer identification [81]. However, a single WCE examination generates thousands of video images, which require physicians to spend a considerable amount of time interpreting them, significantly increasing the diagnostic burden. Therefore, establishing XAI models for automated image analysis can greatly save time and energy and improve the diagnostic speed. Table 4 summarizes XAI applications and visual explanation methods used in WCE.

In the domain of gastrointestinal bleeding detection, Malhi et al. [82] constructed a CNN model using 3895 capsule endoscopy images to identify gastrointestinal bleeding. In the validation set, the model’s accuracy reached 97.92%. Meanwhile, to explain the model’s prediction results, the article selected the LIME method, which has highly reliable results, relatively low computational cost, and model-agnostic nature, to mark the boundaries of gastric bleeding areas [87]. In terms of identifying gastrointestinal ulcers, Wang et al. [83] constructed an ulcer recognition network HAnet-34 (480) with a hyperconnection architecture (HAnet) by using 1157 ulcer videos and 259 normal videos from WCE. The model overall test accuracy is 92.05%. Additionally, they used CAM to display relatively important parts in the images and provide target location information to visualize the model results. For other digestive system diseases, XAI in the form of heatmaps has also been extensively studied. For example, Mukhtorov et al. [84] trained a ResNet152-based CNN model by using an open-source database containing 8000 capsule endoscopy images to identify gastrointestinal diseases. They used the Grad-CAM method to display the image regions that contributed the most to a given classification decision, thereby highlighting the model’s interpretability. On the validation set, the model’s accuracy was as high as 93.46%. Mascarenhas et al. [85] constructed a CNN-based model utilizing colon capsule endoscopy images to automatically detect protruding lesions in the colonic lumen. The model exhibits a high level of performance and excellent image processing capabilities, with both outstanding recognition accuracy and processing speed. The Grad-CAM method generates heatmaps to emphasize regions of diagnostic significance relevant to the prediction of lesions. However, this model was limited to identifying protruding lesions without differentiating their specific types. Furthermore, Nadimi et al. [86] constructed a CNN model trained on 11,300 data-augmented wireless colon capsule endoscopy images to automatically identify colorectal polyps. The model achieved an accuracy of 98%, a sensitivity of 98.1%, and a specificity of 96.3%, which outperformed all the latest results. Additionally, activation maps were used to display the parts that contributed significantly to the results.

## 5. Discussion

Currently, XAI is developing rapidly in the field of digestive endoscopy, with more and more XAI methods emerging. Among them, visual explanation methods employed in medical image analysis are widely applied owing to their plug-and-play characteristics and easily accessible open-source implementations. Therefore, this review outlines the latest progress of XAI based on visual explanation in the field of digestive endoscopy image analysis. Notably, approximately 39% of studies used Grad-CAM, about 22.5% used CAM, and CAM-based cases constituted the majority of the investigations analyzed. A potential reason for this prevalence could be the excellent capabilities of Grad-CAM and CAM in interpreting image analysis models, such as high reliability, high efficiency, and intuitive interpretability. This section will systematically explore the core value of current progress, existing limitations, and future development directions.

Black-box models used to be the primary obstacle hindering the clinical implementation of AI. For clinicians, XAI methods based on visual explanations such as Grad-CAM and LIME visualize lesion-focused regions (such as the morphology of gastrointestinal glandular duct openings, abnormal microvessels, and the abnormal echogenic area of pancreatic lesions, etc.), enabling doctors to intuitively verify the decision-making logic of AI. This visualization has significantly enhanced physicians’ trust and acceptance of AI-assisted diagnostic results, which is crucial in high-risk medical fields. Furthermore, XAI methodologies not only bolster clinical confidence but also serve as educational tools. Physicians, particularly junior physicians, may learn from the model’s decision-making process, which can improve diagnostic accuracy to a certain extent and help discover new imaging markers. Finally, XAI may help physicians identify areas where the model may have focused incorrectly or failed to focus sufficiently, providing insights to refine and improve model performance [88].

However, in the investigation of XAI applications, this study also found some limitations and challenges. First of all, many studies utilize datasets that are relatively small, lack external validation cohorts, or exhibit unbalanced sample distributions among different disease types, so that the quality and quantity of the data are not guaranteed. In ML, a model’s performance is strongly influenced by both the adequacy and representativeness of its training dataset. Poor data quality and insufficient data can affect both the predictive accuracy and the interpretability of the models. Secondly, the types of data applied in most articles are relatively limited in diversity, which may restrict the diagnostic performance of the models. Thirdly, in this survey, most of the XAI articles based on visual explanations provide interpretability by emphasizing regions that critically influence the prediction outcomes, which offers a certain degree of acceptance and credibility for the model results. However, most articles only use XAI methods and lack evaluations of these XAI methods. We cannot determine whether the explanations generated by the models are correct, which makes it necessary to be cautious when interpreting high-risk decisions. As Rudin et al. [15] proposed, the explanations provided by some interpretive models may not accurately represent the original models, and some XAI methods may offer meaningless or insufficient details in interpreting the models. For example, the saliency map is regarded as an interpretability tool, which can highlight the pixels that have the greatest impact on the output and weaken the unnecessary parts. It can tell us where the model is focusing, but we do not understand why the highlighted parts are related to the final result, and the diagnostic basis of the model cannot be fully comprehended. Meanwhile, because saliency maps tend to highlight edges, they may provide similar explanations for each analogy. The reliability of explainable methods remains debatable. Finally, the involvements of medical experts in the design and evaluation of XAI frameworks are limited. Many XAI articles focus on the development of ML models and the deployment of XAI methods, lacking the participation of medical experts. This disconnect may result in XAI applications failing to meet the actual needs of clinicians.

Regarding the future development of XAI, we believe that it will play a crucial role in the auxiliary identification and diagnosis of diseases, as it serves to elucidate the decision-making processes of AI, thereby fostering greater confidence among medical practitioners. To tackle current limitations, we consider that, first of all, in terms of data collection and processing, we should collect multi-center datasets as much as possible, strengthen the utilization of data from public databases, and reasonably employ technologies such as data augmentation and transfer learning to cope with the challenges posed by small datasets. Meanwhile, methods such as denoising and reasonable annotation should be adopted to improve the quality of image data. Secondly, we can integrate multimodal data, such as different types of image data or combining image and non-image data, to build models. This will not only further improve the diagnostic performance of the models but also enhance the clinical acceptance of AI technologies. Thirdly, in the process of data collection and processing, the data quality can be publicly reported to help identify potential inaccuracies or biases, and relevant standards should be formulated to protect user information so as to ensure data security [89,90]. Fourthly, attention should be paid to program resource constraints when developing XAI algorithms. For instance, real-time reporting systems require strong computing and processing capabilities to respond quickly. Therefore, challenges such as high computing demand should be considered during the model development process, and resource-intensive explanation methods should be used cautiously [89]. Fifth, regarding the evaluation methods of XAI, there is currently no universally accepted evaluation standard. Doshi-Velez and Kim [91] once proposed a method for evaluating interpretability, which conducts the evaluation from three levels: application-based evaluation, human-based evaluation, and function-based evaluation. However, the evaluation of XAI methods remains a relatively young research field at present. It is still important to involve experts in the evaluation of XAI results. Developing standards for assessing the interpretability of different models will facilitate the application and development of clinical XAI. Meanwhile, provided that diagnostic efficacy remains comparable, more inherently interpretable models can be constructed in the future, as they seem to be more reasonable when dealing with high-risk decisions. For example, Dong et al. [92] developed a high-performance explainable AI system for diagnosing early gastric tumors based on feature extraction and multi-feature fitting. The result interface intuitively displays 6 characteristics of lesions and final diagnosis results, which significantly improves the transparency of the model. Sixth, incorporating multiple interpretability methods, such as textual explanations and case-based explanations, may yield more comprehensive and accurate explanations in future research. Finally, healthcare professionals ought to play an integral role in the conception and engineering of XAI systems to realize human–machine interaction. While trained XAI models can provide explainable results to support physicians in making correct decisions, this is not sufficient to fully rely on algorithms for medical diagnosis. Clinicians should also provide medical knowledge to guide the design and modification of AI algorithms. By promoting human–machine interaction, XAI models can achieve more successful application in the medical field.

We made great efforts to include as many relevant articles as possible in our study. However, specific visual explanation methods are sometimes not mentioned in the titles or keywords of papers. Therefore, there may be omissions of relevant literature during the review process. Meanwhile, this review only searched the PubMed database. Given the differences in literature inclusion criteria among various databases, some potentially relevant studies may not have been included in this article. Finally, in this review, XAI refers to the relevant research and applications that use XAI methods to explain the output of original models, and does not include models that are inherently interpretable. This serves as a crucial premise for our subsequent elaboration and analysis in the article.

## 6. Conclusions

This article summarizes the clinical applications of XAI employing visual explanations in digestive endoscopic image analysis. Structurally, we first introduce the definition and classification of XAI; secondly, we discuss several commonly used XAI methods based on visual explanations and elaborate on their applications in different digestive endoscopies, respectively; ultimately, we summarize the value, limitations and future development prospects of XAI. We put forward our own insights into the current situation of XAI based on visual explanations in the field of digestive endoscopic image analysis. In the future, in the field of medical image analysis an increasing number of XAIs will emerge to effectively achieve clinical translation. We believe that clinicians should participate in the development, design, and use of models to guide the design of XAIs that conform to clinical workflows and meets clinical needs, so as to give full play to the auxiliary role of XAIs in clinical diagnosis and treatment.

## Figures and Tables

**Figure 1 bioengineering-12-01058-f001:**
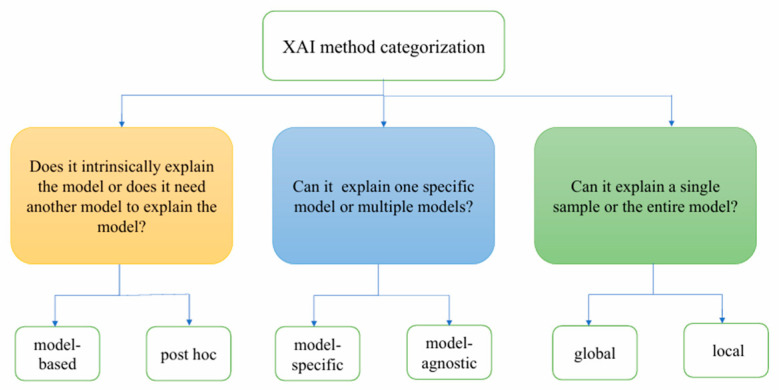
Explainable artificial intelligence method categorization.

**Figure 2 bioengineering-12-01058-f002:**
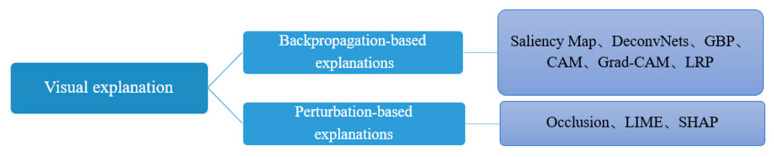
Visual explanation methods categorization.

**Figure 3 bioengineering-12-01058-f003:**
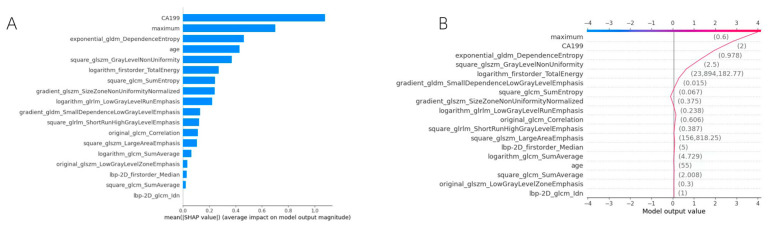
Global explanation and local explanation provided by SHAP. (**A**) The SHAP bar chart is sorted by the mean absolute SHAP values of features to provide global explanation. (**B**) The SHAP decision plot calculates the magnitude and direction of each feature’s contribution to the prediction result in an individual sample to achieve local explanation.

**Table 1 bioengineering-12-01058-t001:** A Summary of Representative Studies on XAI in Esophagogastroduodenoscopy examination.

Types of Digestive Endoscopy	Author	Year	Aim	Main Visual XAI Method
Esophagogastroduodenoscopy	Ge et al. [39]	2023	Diagnosis of Los Angeles classification of GERD	Grad-CAM
	De Souza et al. [40]	2021	Identification of BE and esophageal adenocarcinoma	Saliency map, GBP, etc.
	García-Peraza-Herrera et al. [41]	2020	Classification of IPCLs	CAM
	Everson et al. [42]	2021	Classification of IPCLs	CAM
	Ueyama et al. [43]	2021	Diagnosis of EGC	Grad-CAM
	Hu et al. [44]	2021	Diagnosis of EGC	Grad-CAM
	Cho et al. [45]	2020	Prediction of submucosal invasion for gastric neoplasms	CAM

BE, Barrett’s esophagus; CAM, Class Activation Mapping; EGC, early gastric cancer; GBP, Guided Back Propagation; GERD, gastroesophageal reflux disease; Grad-CAM, Gradient-Weighted Class Activation Mapping; IPCLs, intraepithelial capillary loops; XAI, explainable artificial intelligence.

**Table 2 bioengineering-12-01058-t002:** A Summary of Representative Studies on XAI in Colonoscopy examination.

Types of Digestive Endoscopy	Author	Year	Aim	Main Visual XAI Method
Colonoscopy	Chen et al. [54]	2024	Evaluation of key quality indicators for colonoscopy	Grad-CAM, Guided Grad-CAM, SHAP
	Wickstrom et al. [55]	2020	Polyp recognition	GBP
	Jin et al. [56]	2020	Polyp classification	Grad-CAM
	Zhang et al. [57]	2024	Polyp classification	Grad-CAM
	Choi et al. [58]	2020	Polyp classification	CAM
	Chierici et al. [59]	2022	Identification of CD and UC	GBP
	Sutton et al. [60]	2022	Diagnosis and grading of UC	Grad-CAM
	Weng et al. [61]	2022	Differentiation of CD and ITB	SHAP
	Gabralla et al. [62]	2023	Identification of colon cancer	Grad-CAM
	Binzagr F et al. [63]	2024	Classification of gastrointestinal cancer	SHAP
	Auzine MM et al. [64]	2024	Classification of gastrointestinal cancer	SHAP, LIME
	Dahan et al. [65]	2025	Detection of gastrointestinal disease	Grad-CAM
	Gideon et al. [66]	2025	Detection of gastrointestinal disease	Grad-CAM, SHAP

CAM, Class Activation Mapping; CD, Crohn’s disease; GBP, Guided Back Propagation; Grad-CAM, Gradient-Weighted Class Activation Mapping; ITB, intestinal tuberculosis; LIME, Local Interpretable Model-agnostic Explanations; SHAP, SHapley Additive exPlanations; UC, ulcerative colitis; XAI, explainable artificial intelligence.

**Table 3 bioengineering-12-01058-t003:** A Summary of Representative Studies on XAI in Endoscopic Ultrasonography examination.

Types of Digestive Endoscopy	Author	Year	Aim	Main Visual XAI Method
Endoscopic Ultrasonography	Gu et al. [72]	2023	Diagnosis of PDAC	Grad-CAM
	Yi et al. [73]	2025	Identification of PDAC and PNETs	Grad-CAM, SHAP
	Marya et al. [74]	2021	Diagnosis of pancreatic diseases	Occlusion
	Mo et al. [75]	2025	Prediction of PNETs pathological grading	SHAP
	Cai et al. [76]	2025	Diagnosis of pancreatic diseases	SHAP
	Cui et al. [77]	2024	Diagnosis of pancreatic diseases	Grad-CAM, SHAP
	Liang et al. [78]	2025	Preoperative risk prediction of GISTs	SHAP
	Liu et al. [79]	2022	Identification of the lesion invasion depth and lesion source of esophageal submucosal tumors	CAM
	Uema et al. [80]	2024	Identification of the invasion depth of EGC	CAM

CAM, Class Activation Mapping; EGC, early gastric cancer; GISTs, gastrointestinal stromal tumors; Grad-CAM, Gradient-Weighted Class Activation Mapping; PDAC, pancreatic ductal adenocarcinoma; PNETs, pancreatic neuro endocrine tumors; SHAP, SHapley Additive exPlanations; XAI, explainable artificial intelligence.

**Table 4 bioengineering-12-01058-t004:** A Summary of Representative Studies on XAI in Wireless Capsule Endoscopy examination.

Types of Digestive Endoscopy	Author	Year	Aim	Main Visual XAI Method
Wireless Capsule Endoscopy	Malhi et al. [82]	2019	Detection of gastrointestinal bleeding	LIME
	Wang et al. [83]	2019	Recognition of peptic ulcer	CAM
	Mukhtorov et al. [84]	2023	Detection of gastrointestinal disease	Grad-CAM
	Mascarenhas et al. [85]	2022	Diagnosis of colonic protruding lesions	Grad-CAM
	Nadimi et al. [86]	2020	Detection of colorectal polyps	Saliency map

CAM, Class Activation Mapping; Grad-CAM, Gradient-Weighted Class Activation Mapping; LIME, Local Interpretable Model-agnostic Explanations; XAI, explainable artificial intelligence.

## Data Availability

Not applicable.

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
