# Peer review of "Application of Explainable Artificial Intelligence Based on Visual Explanation in Digestive Endoscopy"

_bioengineering, 2025, doi:10.3390/bioengineering12101058_

Round 1
Reviewer 1 Report
Comments and Suggestions for Authors
This manuscript presents a topical review on the use of Explainable AI (XAI) in digestive endoscopy. While the subject is timely and relevant, I have several concerns that should be addressed to improve the quality and clarity of the paper:
1. Title: The word "endoscopy" should be written as one word. Additionally, the phrase “Unveiling … black box” can be removed for conciseness and clarity.
2. Introduction: The authors have overlooked some recent and relevant publications that provide foundational background on AI applications in medical imaging, particularly for therapy and disease diagnosis. Notable examples include:
• Algorithms, 2025, 18(3), 156. https://doi.org/10.3390/a18030156
These references should be incorporated to strengthen the introduction.
3. Section 2: It would be beneficial to include a comparative table outlining the differences between XAI and non-XAI approaches. This should be followed by a discussion highlighting the specific advantages of XAI in the context of digestive endoscopy.
4. Section 4: I recommend adding one or two illustrative figures demonstrating XAI applications. These can be adapted from other imaging studies that utilize XAI, provided proper attribution is given.
5. Limitations: The manuscript should include a dedicated paragraph or section discussing the limitations and assumptions of the review, to provide a balanced perspective.
6. GAI Contribution: If any part of this work was generated or assisted by Generative AI (GAI), the authors should explicitly declare this at the end of the manuscript.
Author Response
Thank you very much for taking the time to review this manuscript. According to your valuable suggestions, we have made extensive corrections to our previous draft. The reviewer comments are laid out below and specific concerns have been numbered. We have used track changes in the revised manuscript, with all changes highlighted in red.
Comments 1: Title: The word "endoscopy" should be written as one word. Additionally, the phrase “Unveiling … black box” can be removed for conciseness and clarity.
Response 1:
Thank you for pointing this out. We agree with this comment. Therefore, we have merged the “endoscopy” into one word and removed the “Unveiling … black box”.
[The revised title of the article is: Application of explainable artificial intelligence based on visual explanation in digestive endoscopy]
Comments 2: Introduction: The authors have overlooked some recent and relevant publications that provide foundational background on AI applications in medical imaging, particularly for therapy and disease diagnosis. Notable examples include:
• Algorithms, 2025, 18(3), 156. https://doi.org/10.3390/a18030156IF: 2.1 Q2 These references should be incorporated to strengthen the introduction.
Response 2:
Thank you for your valuable comments. We have carefully reviewed the relevant literature and added background information on the application of AI models in medical image analysis. (lines 54-57)
[as follows: Currently, AI is increasingly being used in medical imaging. By training on large amounts of medical image data, it enables the automatic detection and diagnosis of lesions, which helps clinicians make faster and more informed decisions and improves outcomes for patients.]
Comments 3: Section 2: It would be beneficial to include a comparative table outlining the differences between XAI and non-XAI approaches. This should be followed by a discussion highlighting the specific advantages of XAI in the context of digestive endoscopy.
Response 3:
Thank you for your valuable comments. Compared with non-XAI, XAI not only provides prediction results but also offers explanations for the output results. It features higher model transparency, making it easier to audit the ethics and fairness of the AI model, and users also have greater trust in its results. Meanwhile, when the model makes errors in its results, understanding how the model makes decisions can help with model modifications (as shown in the table below).
However, in the second paragraph of the discussion section (lines 641-653), we have also provided detailed examples to illustrate the advantages of XAI in clinical applications (compared with non-XAI). Specifically, in the field of digestive endoscopy image analysis, XAI can highlight features such as the opening of gastrointestinal glands, abnormal microvascular morphology, and abnormal pancreatic echo areas, thereby providing interpretable analysis.
Therefore, considering the above reasons, we have not added the following table to this article.
|
XAI |
Non-XAI |
|
|
Operating Principle |
Provide the reasons and basis for the model to make decisions. |
Provide only the prediction results. |
|
Model Transparency |
High |
Low |
|
Model Trustworthiness |
High |
Low |
|
Model Debuggability |
Easy |
Difficult |
|
Ethics and Fairness |
Easy to audit |
Difficult to audit |
Comments 4: Section 4: I recommend adding one or two illustrative figures demonstrating XAI applications. These can be adapted from other imaging studies that utilize XAI, provided proper attribution is given.
Response 4:
Thank you for your valuable comments. We have added one figure (Figure 3, line 550) in the article to illustrate the clinical application of the XAI method (SHAP analysis).
SHAP analysis is a common XAI method that can provide both global explanations and local explanations for models. In figure 3, (A) The SHAP bar chart is sorted by the mean absolute SHAP values of features to provide global explanation. The results show that CA199 is the most important feature affecting the model output (pancreatic malignant lesions vs. pancreatic benign lesions). (B) The SHAP decision plot calculates the magnitude and direction of each feature's contribution to the prediction result in an individual sample to achieve local explanation. The results show that for a patient with benign pancreatic lesion, the size of the patient's lesion is the most important feature affecting the model output.
Comments 5: Limitations: The manuscript should include a dedicated paragraph or section discussing the limitations and assumptions of the review, to provide a balanced perspective.
Response 5: Thank you for your valuable comments. We have added a dedicated paragraph in the article to explain the limitations and assumptions of this review. (lines 723-731)
[The added content is as follows: We made great efforts to include as many relevant articles as possible in our study. However, specific visual explanation methods are sometimes not mentioned in the titles or keywords of papers. Therefore, there may be omissions of relevant literature during the review process. Meanwhile, this review only searched the PubMed data-base. Given the differences in literature inclusion criteria among various databases, some potentially relevant studies may not have been included in this article. Finally, in this review, XAI refers to the relevant research and applications that use XAI methods to explain the output of original models, and does not include models that are inherently interpretable. This serves as a crucial premise for our subsequent elaboration and analysis in the article.]
Comments 6: GAI Contribution: If any part of this work was generated or assisted by Generative AI (GAI), the authors should explicitly declare this at the end of the manuscript.
Response 6: Thank you for pointing this out. We confirm that no Generative AI technologies were used in any stage of the preparation or writing of this manuscript.
Reviewer 2 Report
Comments and Suggestions for Authors
The manuscript aims to review the deep learning models and decision explanation methods for image classification in the field of medical diagnostics of cancers in endoscopy, colonoscopy, ultrasonography and capsule endoscopy. The authors decided to focus on visual explanations and explore Grad-CAM, Saliency Maps and SHAP values. The text is well written and easy to follow, a lot of important features are discussed. Unfortunately, the paper lacks visual examples. Also, I think, the title is not precise as it seems to promise something new to unveil some mystery.
Major concerns
Some numerical results are to be added to the Abstract, some strong conclusion may be also added.
lines 35-38 This view to ML is too "popular", not scientific. ML does require programing, computers are made of metal and plastic and can't learn. Parameters of mathematical fucntions of the model are "learnt" - deduced from data - using algorithms that were programmed. The mathematical model is called "machine" for short.
The review must firstly describe the databases and criteria for source selection. The dynamics for the number of papers that satisfy these criteria is to be presented.
The comparisons with previous reviews is to be done.
Lines 77-85 repeat the contents of introduction
Sections 2.2.1 and 2.2.2 have examples so 2.2.3 also needs an example.
Tables have to be placed closer to the first reference to it.
The use of transformers is not discussed. The quick googling gives:
Chen, J., Wang, G., Zhou, J. et al. AI support for colonoscopy quality control using CNN and transformer architectures. BMC Gastroenterol 24, 257 (2024). https://doi.org/10.1186/s12876-024-03354-0
Siddhartha Kapuria, Naruhiko Ikoma, Sandeep Chinchali, Farshid Alambeigi, Synthetic data-augmented explainable Vision Transformer for colorectal cancer diagnosis via surface tactile imaging, Engineering Applications of Artificial Intelligence, Volume 151, 2025, 110633, ISSN 0952-1976, https://doi.org/10.1016/j.engappai.2025.110633.
Gabralla, L. A., Hussien, A. M., AlMohimeed, A., Saleh, H., Alsekait, D. M., El-Sappagh, S., Ali, A. A., & Refaat Hassan, M. (2023). Automated Diagnosis for Colon Cancer Diseases Using Stacking Transformer Models and Explainable Artificial Intelligence. Diagnostics, 13(18), 2939. https://doi.org/10.3390/diagnostics13182939
etc
line 650 The paper describes model training and validation, the clinical application require adoption in practice by doctors that are not the developers of the models but the users.
Minor issues
line 33 "is consist"?
The literature cited is numbered twice.
Author Response
Thank you very much for taking the time to review this manuscript. According to your valuable suggestions, we have made extensive corrections to our previous draft. The reviewer comments are laid out below and specific concerns have been numbered. We have used track changes in the revised manuscript, with all changes highlighted in red.
Comments 1: Some numerical results are to be added to the Abstract, some strong conclusion may be also added.
Response 1:
Thank you for pointing this out. We agree with this comment. Therefore, we have added some numerical results (lines 20-22) and strong conclusion in the Abstract (lines 30-33).
[numerical results: The review systematically reviews 34 articles (7 articles in esophagogastroduodenoscopy, 13 articles in colonoscopy, 9 articles in endoscopic ultrasonography, and 5 articles in wireless capsule endoscopy)]
[strong conclusion: This study provides a theoretical basis for constructing a trustworthy and transparent AI-assisted digestive endoscopy diagnosis and treatment system, and promotes the implementation and application of XAI in clinical practice.]
Comments 2: lines 35-38 This view to ML is too "popular", not scientific. ML does require programing, computers are made of metal and plastic and can't learn. Parameters of mathematical fucntions of the model are "learnt" - deduced from data - using algorithms that were programmed. The mathematical model is called "machine" for short.
Response 2: Thank you for your valuable comments. We have provided a more scientific explanation of the concept of ML, and we point out that: Machine Learning (ML) is a subset of AI. Its core lies in constructing a mathematical model through algorithms and optimizing the model's parameters using data, thereby enabling it to make accurate predictions or decisions on unknown data. (lines 40-43).
Comments 3: The review must firstly describe the databases and criteria for source selection. The dynamics for the number of papers that satisfy these criteria is to be presented.
Response 3: Thank you for pointing this out. We agree with this comment. We have added the search strategy and inclusion criteria to make it more in line with the requirements for a review. (lines 272-283)
[search strategy and inclusion criteria: We searched articles published in the PubMed database from 2015 to 2025 to identify the current applications of XAI based on visual explanation in the analysis of digestive endoscopic images. The search terms are as follows: ((“explainable artificial intelligence” OR “interpretable artificial intelligence” OR “artificial intelligence” OR “AI” OR “machine learning” OR “deep learning” OR “convolutional neural network”)) AND (endoscopy). The inclusion criteria are as follows: (1) The XAI models are applied to digestive endoscopic image analysis; (2) The article adopts specific visual explanation methods to explore the interpretability of AI models; (3) Complete article information is accessible. We also included key articles identified through manual searching or reference lists. Then, the above articles were manually screened and evaluated again, and a total of 34 eligible articles were finally included.]
Comments 4: The comparisons with previous reviews is to be done.
Response 4: Thank you for your valuable comments. In the introduction section, we have added a comparison with previous related articles. (lines 79-93)
[comparison with previous related articles: Van der Velden et al. included 223 relevant studies to comprehensively analyze the application of XAI methods in medical image analysis. They discussed and compared the advantages and disadvantages of various XAI methods, and put forward their unique insights into the future development of XAI. The results showed that visual explanation is the most commonly used interpretability method in medical image analysis. Salih et al. included 32 articles to conduct a retrospective analysis of the application of XAI in various cardiac imaging methods. Among them, the visual ex-planation method achieves model interpretability by visualizing relevant features, ac-counting for approximately half of the article. Similarly, Qian et al.included 56 articles to study the application of XAI in the analysis of MRI images of different human body parts, and identified the evaluation metrics for XAI methods. The results showed that XAI based on visual explanation accounted for more than 60%. The above studies highlight the widespread clinical application of XAI based on visual explanation. However, there are still relatively few reviews on XAI, especially XAI based on visual explanation, in the field of digestive endoscopic image analysis.]
Comments 5: Lines 77-85 repeat the contents of introduction
Response 5: Thank you for your reminder. We have deleted the repeat parts. (“lines 77-85” / lines 107-115 have been deleted)
Comments 6: Sections 2.2.1 and 2.2.2 have examples so 2.2.3 also needs an example.
Response 6: Thank you for your valuable comments. We have added an example (SHAP analysis) in section 2.2.3 to illustrate global explanations and local explanations. (lines 160-165)
[example: For example, SHapley Additive exPlanations (SHAP)analysis can not only provide global explanations to reveal the ranking of important features related to overall pre-diction results, but also provide important features associated with the result of individual sample to provide local explanations.]
Comments 7: Tables have to be placed closer to the first reference to it.
Response 7: Thank you for your reminder. We have adjusted the position of the table to place it closer to where it is first cited.
Comments 8: The use of transformers is not discussed. The quick googling gives:
Chen, J., Wang, G., Zhou, J. et al. AI support for colonoscopy quality control using CNN and transformer architectures. BMC Gastroenterol 24, 257 (2024). https://doi.org/10.1186/s12876-024-03354-0IF: 2.6 Q2
Siddhartha Kapuria, Naruhiko Ikoma, Sandeep Chinchali, Farshid Alambeigi, Synthetic data-augmented explainable Vision Transformer for colorectal cancer diagnosis via surface tactile imaging, Engineering Applications of Artificial Intelligence, Volume 151, 2025, 110633, ISSN 0952-1976, https://doi.org/10.1016/j.engappai.2025.110633IF: 8.0 Q1 .
Gabralla, L. A., Hussien, A. M., AlMohimeed, A., Saleh, H., Alsekait, D. M., El-Sappagh, S., Ali, A. A., & Refaat Hassan, M. (2023). Automated Diagnosis for Colon Cancer Diseases Using Stacking Transformer Models and Explainable Artificial Intelligence. Diagnostics, 13(18), 2939. https://doi.org/10.3390/diagnostics13182939IF: 3.3 Q1
Etc
Response 8: Thank you for your valuable comments. We have supplemented three examples of the application of Transformers in the article to enrich the content of the article. (lines 395-401、lines 465-470、lines 478-483).
Comments 9: line 650 The paper describes model training and validation, the clinical application require adoption in practice by doctors that are not the developers of the models but the users.
Response 9: Thank you for your valuable comments. We have revised the final part of the conclusion. (lines 744-747)
[ as follows: We believe that clinicians should participate in the development, design, and use of models, guide the design of XAI that conforms to clinical workflows and meets clinical needs, so as to give full play to the auxiliary role of XAI in clinical diagnosis and treatment.]
Comments 10: line 33 "is consist"?
Response 10: Thank you for pointing this out. This is an error. Since we have redefined ML, "is consist" has been removed.
Comments 11: The literature cited is numbered twice.
Response 11: Thank you for your reminder. We have adjusted the references to meet the requirements of the article.
Round 2
Reviewer 1 Report
Comments and Suggestions for Authors
I am satisfied with the additional contents and corrections from the authors as per my comments. The quality and presentation of the manuscript are improved.
Reviewer 2 Report
Comments and Suggestions for Authors
The authors took into account almost all my suggestions. So though the number of reviewed articles is not that big and the selected publications numbers for different years are not given in the text I have only minor comments regarding appearance - the text doesn't need to be formatted in bold, eg lines 702-13, and the title 6. Conclusions is to be placed to a new line.